# Heterogeneous Multi-Material Flexible Piezoresistive Sensor with High Sensitivity and Wide Measurement Range

**DOI:** 10.3390/mi14040716

**Published:** 2023-03-23

**Authors:** Tingting Yu, Yebo Tao, Yali Wu, Dongguang Zhang, Jiayi Yang, Gang Ge

**Affiliations:** 1School of Aerospace Science and Technology, Xidian University, Xi’an 710071, China; 2Intelligent Manufacturing College, Jiaxing Vocational & Technical College, Jiaxing 314036, China; 3College of Mechanical and Vehicle Engineering, Taiyuan University of Technology, Taiyuan 030024, China; 4College of Computer Science and Technology, Xi’an University of Science and Technology, Xi’an 710054, China; 5Department of Electrical and Computer Engineering, National University of Singapore, Singapore 117583, Singapore

**Keywords:** soft sensors, pressure sensors, stress sensors, high dynamics range, graphene

## Abstract

Flexible piezoresistive sensors (FPSs) have the advantages of compact structure, convenient signal acquisition and fast dynamic response; they are widely used in motion detection, wearable electronic devices and electronic skins. FPSs accomplish the measurement of stresses through piezoresistive material (PM). However, FPSs based on a single PM cannot achieve high sensitivity and wide measurement range simultaneously. To solve this problem, a heterogeneous multi-material flexible piezoresistive sensor (HMFPS) with high sensitivity and a wide measurement range is proposed. The HMFPS consists of a graphene foam (GF), a PDMS layer and an interdigital electrode. Among them, the GF serves as a sensing layer, providing high sensitivity, and the PDMS serves as a supporting layer, providing a large measurement range. The influence and principle of the heterogeneous multi-material (HM) on the piezoresistivity were investigated by comparing the three HMFPS with different sizes. The HM proved to be an effective way to produce flexible sensors with high sensitivity and a wide measurement range. The HMFPS-10 has a sensitivity of 0.695 kPa^−1^, a measurement range of 0–14,122 kPa, fast response/recovery (83 ms and 166 ms) and excellent stability (2000 cycles). In addition, the potential application of the HMFPS-10 in human motion monitoring was demonstrated.

## 1. Introduction

Flexible stress sensors are flexible electronics that conform to curved surfaces and convert stress into electrical signals [1,2,3,4]; they are widely used in motion detection [5,6,7], wearable electronics [8,9], human–computer interaction [10,11,12], and electronic skin [13,14,15]. Flexible stress sensors can be classified into piezoresistive [16], capacitive [17,18], piezoelectric [19] and triboelectric [20] sensors according to different sensing mechanisms. Among them, flexible piezoresistive sensors (FPSs) can convert stress into electrical signals through changes in the conductive paths, which have the advantages of simple structure, convenient signal acquisition, fast dynamic response and cost-efficiency [21,22]. Two important parameters of FPSs are sensitivity and measurement range [18,23]. High sensitivity enables the sensor to accurately identify small stress [24], while a wide measurement range is the key to ensuring the applicability of the sensor in various stress scenarios. Therefore, it is of great significance to achieve an FPS with high sensitivity and a wide measurement range [25].

FPSs usually consist of electrodes and the piezoresistive materials (PM), which determine the sensitivity and measurement range of the sensor. There are three typical piezoresistive materials: conductive films (CFs) [26,27], conductive composites (CCs) [28,29] and conductive porous foam (CPFs) [30,31]. CFs are composed of conductive films on flexible substrates [26,27]. Cracks or breaks appear on the surface’s conductive layer after it is stressed, which changes the conductive path and causes a variation in resistance. CFs have the advantage of high sensitivity, while the measurement range of CFs is limited due to the low deformability [32]. CCs are conductive ingredient-doped polymers [33,34]. Appling stress changes the relative position of the conductive filler inside the CCs, resulting the establishment or break of the conductive paths. The change in the conductive paths requires a decent strain, resulting in a large measurement range and a low sensitivity. CPFs are conductive material-coated porous polymers [31,35]. Applying stress deforms the three-dimensional conductive skeleton of the porous polymer, which establishes conductive paths through contact with conductive skeletons. CPFs has better flexibility and piezoresistive properties. Compared with CFs, the three-dimensional porous skeleton can provide a larger measurement range. Comparing with CCs, the three-dimensional porous skeleton has a lower modulus and can provide higher sensitivity.

To improve the sensitivity and measurement range of CPF-based sensors, surface microstructures (such as micropillars [36], domes [37,38], micro-pyramids [39], microgrooves [40], and folds [41]) or gradient porous structures [18,42] are proposed. Although these works improve the sensitivity or measurement range, achieving high sensitivity and wide measurement range simultaneously is still a severe bottleneck that restricts the practical applications of these sensors.

Heterogeneous multi-material (HM) is composed of different materials with continuous or discontinuous structures distribution, and can obtain properties that cannot be achieved by a single material [43,44]. The CPF has a piezoresistivity and lower elastic modulus, while the elastomer has a high elastic modulus. Therefore, combining the CPF and the elastomer to design an HM is a potential solution to achieve high sensitivity and a wide measurement range in one sensor.

To solve the problem, a heterogeneous multi-material flexible piezoresistive sensor (HMFPS) with high sensitivity and a wide measurement range is proposed. The HMFPS consists of a graphene foam (GF) sensing layer, a polydimethylsiloxane (PDMS) supporting layer and an interdigital electrode. Among them, the GF is fixed at the center of the PDMS supporting layer, and the flexible interdigital electrode is placed under the surface of the GF, as shown in Figure 1a. We prepared the HMFPS with different PDMS supporting layer areas, namely HMFPS-3, HMFPS-5, HMFPS-10. The influence of the HM structure on the performance of the sensor and its working principle is investigated, proving that the HM structure is an effective way to improve the piezoresistivity of the FPS. Among them, HMFPS-10 has the best comprehensive performance, with a sensitivity of 0.695 KPa^−1^, a measurement range of 0–14,122 KPa, fast response/recovery (83 ms and 166 ms) and excellent stability (2000 cycles). In addition, the flexible sensor can be applied to human movement monitoring.

## 2. Experimental

### 2.1. Materials

PDMS (Sylgard 184) and curing agent were provided by Dow Corning Co., Ltd. (Midland, MI, USA). Graphene oxide (GO) powder was purchased from Nanjing XFNANO Materials Tech. Co., Ltd. (Nanjing, China). Melamine foams (MF) were obtained from Shenzhen Siyuan Rubber and Plastic Products Factory (Shenzhen, China). Vitamin C (VC) was purchased from Sinopharm Chemical Reagent Co., Ltd. (Shanghai, China). An interdigital electrode with polyimide (PI) substrate was purchased from Rigorous Technology Co., Ltd. (Shenzhen, China). All the materials and chemicals were used as received.

### 2.2. Preparation of HMFPS

The HMFPS consists of a GF sensing layer, a PDMS supporting layer and an interdigital electrode. Among them, the GF is fixed at the center of the PDMS supporting layer, and the flexible interdigital electrode is placed under the surface of the GF, as shown in Figure 1a. The preparation process of the HMFPS is shown in Figure 1b. First, the GFs were prepared by a dip-coating method. MFs (10 mm × 10 mm × 8 mm) were washed in deionized water and ethanol for 10 min then dip-coated in GO solution (2 mg mL^−1^) under vacuum for 10 min. The resulting GOMFs were immersed in vitamin C (VC) solution (12 mg mL^−1^) for 6 h at 90 °C to reduce the GO to rGO. The obtained GFs were washed in deionized water to remove the excess VC, and dried at 80 °C. We prepared five groups of GOMFs with different dip-coating times, respectively. The samples were chemically reduced to prepare the GF, expressed as GF_x_ (where x = 1, 2, 3, 4, and 5). Second, the PDMS supporting layers were prepared. PDMS and curing agent were mixed in a ratio of 10:1, then poured into a Teflon mold after removing bubbles. The mixture was cured at 80 °C for 4 h to obtain a PDMS supporting layer. We prepared three different sizes of PDMS supporting layers with widths of 3 mm, 5 mm, and 10 mm, respectively. Finally, the GF, PDMS supporting layer and interdigital electrode were assembled to obtain the HMFPS. The GF is fixed on the surface of the interdigital electrode using a conductive silver paste. Additionally, the copper wire is fixed to the interdigital electrode by soldering tin. The interdigital electrode enables the sensor to be compact and avoid exposed wire connections, increasing the stability of the sensor. The HMFPSs with three different supporting layer sizes are defined as HMFPS-3, HMFPS-5, and HMFPS-10, respectively.

### 2.3. Characterizations and Measurement

The structures of MF and GF were characterized by scanning electron microscopy (SEM) (Gemini SEM500, Zeiss, Oberkochen, Germany) at an operating voltage of 2 kV. Raman spectrums were recorded using a Raman spectrometer (Jobin Yvon LavRam HR800, Horiba, Japan) with laser excitation of 514 nm.

The mechanical and electrical property tests were conducted on HMFPS-3, HMFPS-5 and HMFPS-10 using a high-precision stress testing machine (TianYuan Test Instrument TY8000, Shanghai, China) and a precision source meter (2450, Keithley, Cleveland, OH, USA), respectively. The sensors were compressed to strain at a constant speed of 20 mm min^−1^, and the loading rate remained at 60 mm min^−1^ for the endurance cycle tests. All the tests were conducted at room temperature.

## 3. Results and Discussion

### 3.1. Structure and Morphology

The HMFPS consists of three parts: the GF sensing layer, the PDMS supporting layer and an interdigital electrode. The GF is 3 mm higher than the PDMS supporting layer, which is fixed at the center of the PDMS supporting layer. The GF is connected to the interdigital electrode by silver paste. Three HMFPSs with different PDMS supporting layer areas (HMFPS-3, HMFPS-5 and HMFPS-10) are prepared. The photographs of the samples are shown in Figure 2a. We took the SEM images to observe the morphology of the GF sensing layer. Figure 2b–d and Figure 2e–g are the SEM images of MF and GF under different magnifications, respectively. The microscopic porous structures of the MF and GF are similar, indicating that the GF prepared by dip-coating method preserves the original pore structure of the MF. In addition, the surface of the MF is flat and smooth, while the surface of the GF presents an obvious fold structure, indicating that graphene sheets are successfully assembled on the MF skeleton.

The porosity of the GF can be expressed as $Q=\frac{V_{p}}{V_{0}}\times 100\%$, where $V_{0}$ represents the volume of GF in its natural state ($V_{0}=0.8\;{\mathrm{cm}}^{-3}$), and $V_{p}$ represents the porous volume of GF). We tested the porosity of the GF by drainage method. The quality of the GF is 6.75 mg. The density of the GF can be calculated as 8.43 kg m^−3^. Additionally, when the GF is full of water, its quality is 635.71 mg. Therefore, the quality of water absorbed by GF pores is 628.96 mg, which is equivalent to the volume of GF pores ($V_{p}=0.629\;{\mathrm{cm}}^{-3}$). Hence, the porosity of the GF is 78.62%.

Appendix A shows the Raman spectra of the GOMF and the GF contain D and G valence bands at 1343 cm^−1^ and 1580 cm^−1^, respectively. The D band is attributed to sp3 defects in the graphene sheet, and the G band is attributed to the E2g phonon mode in the sp2 carbon atoms plane. The ratio of the intensity of the D valence band to the G valence band (I_D_/I_G_) of the GOMF was 0.82. The I_D_/I_G_ of GF was 1.28 after chemical reduction by VC, indicating that the GO sheets attached to MF are efficiently reduced to G.

### 3.2. Sensing Mechanism and Piezoresistivity of the HMFPS

The sensing mechanism of the HMFPS is shown in Figure 3. Applying stress makes the GF skeleton of the HMFPS contact, which decreases the resistance by establishing new conductive paths. The sensitivity of the stress sensor (S) can be defined as $S=\left(\Delta I/I_{0}\right)/P$, where ΔI is the current variation, $I_{0}$ is the original current, and P is the applied stress. Due to the PDMS supporting layer and protruding GF, the piezoresistive effect of HMFPS presents two modes, as shown in Figure 3a. Without stress, the GF is higher than the PDMS supporting layer. With a small stress (<1.8 kPa), the GF above the PDMS supporting layer is compressed, while the PDMS supporting layer is not compressed. At this time, the overall equivalent elastic modulus of the sensor is equal to the GF, leading to a high S. However, with the stress increasing, the GF above the PDMS supporting layer is completely compressed. With more compressive strain, the GF and PDMS supporting layer are compressed simultaneously. In this case, the equivalent elastic modulus of the sensor is approximately equal to the PDMS supporting layer. Compared to only compressed GF, greater stress is required at the same strain, which greatly improves the measurement range of the sensor. In conclusion, the GF above the PDMS supporting layer allows the sensor to sense small stresses (<1.8 kPa). When the PDMS supporting layer is compressed, the sensor can sense large stress (14,122 kPa). Based on the HM design method, the low modulus of the GF can be combined with the high modulus of the PDMS, producing a sensor with high sensitivity and a large measurement range, simultaneously.

To further analyze the piezoresistivity of the sensor, we established the equivalent circuit model of the sensor, as shown in Figure 3b. The total resistance ($R_{total}$) of the HMFPS consists of the GF bulk resistance ($R_{b}$), the interdigital electrode resistance ($R_{f}$) and the contact resistance between the GF and the interdigital electrode ($R_{c}$). The total resistance can be calculated as follows: $R_{total}=R_{b}+R_{f}+R_{c}$. Generally, the $R_{f}$ and $R_{c}$ can be neglected, and the piezoresistivity of the sensor is mainly determined by $R_{b}$. In this study, five groups of GF samples were prepared by changing the dip-coating times. The conductivity of GF increases with the increase in dip-coating times, as shown in Appendix A. The conductivity of the GF increased rapidly for the first three dip-coating times, while it increased slowly for the fourth and fifth dip-coating times. According to the threshold effect, higher sensitivity can be obtained by keeping the conductivity of piezoresistive composites near the threshold value. Therefore, in order to obtain better pressure-sensitive characteristics, GF_3_ foam samples with the largest change in conductivity should be selected. The conductivity of GF_3_ is about 0.086 S m^−1^. In the follow-up studies of this experiment, GF_3_ samples were used. $R_{b}$ depends on the number of conductive paths of the GF and can be equivalent to a sliding rheostat. Without stress, HMFPS remains in its original state, with a resistance of $R_{0}$. Applying stress compresses the GF, resulting in a deformation (Δh). The $R_{b}$ of the GF decreases according to $R_{b}\propto 1/\Delta h$. The sensitivity and measurement range of the sensor are determined by the relationship between the Δh and the P, which can be expressed as E=P/Δh (where E represents elastic modulus of the sensor). Within a certain stress range, the applied stress only works on the GF, and the E of the sensor is small. The smaller P can produce a larger Δh, indicating a large variation rate of the $R_{b}$, so that the sensor has high sensitivity. Beyond a certain range, the existence of the PDMS supporting layer increases the E of the sensor, which generates a certain amount of the Δh that requires a larger P. The variation rate of the $R_{b}$ decreases, increasing the measurement range of the sensor.

The sensing mechanism of the HMFPS is shown in Figure 3c. The entire sensing range can be divided into two stages, which are the low stress range and the high stress range. In the original state, graphene is uniformly covered on the MF surface, forming a stable conductive network. Applying stress contacts the adjacent skeleton of the GF, establishing new conductive paths and decreasing the $R_{b}$. With the increase in the stress, the foam skeletons further contact each other to establish conductive paths, and the $R_{b}$ reduced. The changes in the entire sensing process are attributed to the “contact effects”.

### 3.3. Mechanical Properties of the HMFPS

The size of the PDMS supporting layer determines the equivalent elastic modulus of the sensor, thereby affecting the sensitivity and measurement range. Figure 4a–d are the compressive stress–strain curves of the GF, the HMFPS-3, the HMFPS-5, and the HMFPS-10 under 40%, 60%, and 80% compressive strain. 

Figure 4e shows the compressive stress–strain curves of the GF, the HMFPS-3, the HMFPS-5, the HMFPS-10 and the PDMS. The stress required at the same strain for the HMFPS is sharply increased with the presence of the PDMS supporting layer. The stress of the HMFPS increases nonlinearly with the strain greater than 37.5%. When the strain is less than 37.5%, only the GF is compressed. Due to the lower elastic modulus of the GF, the sensor exhibits a lower elastic modulus (0.0633 kPa). When the strain is greater than 37.5%, the GF and the PDMS supporting layer need to be compressed simultaneously, which equivalently increases the stress required for the deformation of the GF. Figure 4f compares the E of the GF, the HMFPS-3, the HMFPS-5, the HMFPS-10 and the PDMS when the strain is less than 37.5% and greater than 37.5%. When the strain is less than 37.5%, the E of the sample decreases with the increase in the PDMS supporting layer area, which is because the compressed area of the sample (S) is equivalent to the size of the entire sensor during the experiment, that is, $S_{GF}<S_{\mathrm{HMFPS}-3}<S_{\mathrm{HMFPS}-5}<S_{\mathrm{HMFPS}-10}$. The applied stress (P) can be defined as P=F/S, where F represents the applied stress, and S represents the area of the applied stress. The same F ($F_{GF}=F_{\mathrm{HMFPS}-3}=F_{\mathrm{HMFPS}-5}=F_{\mathrm{HMFPS}-10}$) leads to the same Δh ($\Delta h_{GF}=\Delta h_{\mathrm{HMFPS}-3}=\Delta h_{\mathrm{HMFPS}-5}=\Delta h_{\mathrm{HMFPS}-10}$), resulting in $P_{GF}>P_{\mathrm{HMFPS}-3}>P_{\mathrm{HMFPS}-5}>P_{\mathrm{HMFPS}-10}$. Additionally, as E=P/Δh, it can be seen that $E_{GF}>E_{\mathrm{HMFPS}-3}>E_{\mathrm{HMFPS}-5}>E_{\mathrm{HMFPS}-10}$. During this process, the sensitivity of the sensor is equal to that of the GF, which enables the small stress (<1.8 kPa) sensation. When the strain is greater than 37.5%, the E of the sample increases with the increase in the PDMS supporting layer area, which increases the measurement range of the sensor. Therefore, the HMFPS has a larger measurement range compared with the GF, and has high sensitivity compared with the CC-based sensors.

### 3.4. Piezoresistivity of the HMFPS

The current response curves of the GF, the HMFPS-3, the HMFPS-5 and the HMFPS-10 under different stresses are shown in Figure 5a,b. The sensitivity (S) of the HMFPS can be defined as $S=(\frac{\Delta I}{I_{0}})/P$, where ΔI is the current variation under a certain stress, $I_{0}$ is the original current of the GF, and P is the applied stress. Under small stress (<1.8 kPa), the sensor has ultra-high sensitivity of 0.695 kPa^−1^ due to the compression of only the GF sensing layer. With the increase in the stress, the GF sensing layer and the PDMS supporting layer are compressed simultaneously. The required stress increases with the current increases, which is affected by the PDMS supporting layer. The sensitivity of the HMFPS decreases, while the measurement range of the increases. This is consistent with the sensor mechanism of the sensor. Figure 5c shows the sensitivity and measurement range of the GF, the HMFPS-3, the HMFPS-5 and the HMFPS-10. In the small stress range (<1.8 kPa), the sensitivity of the HMFPS is 0.695 kPa^−1^. HMFPS-10 has a wide stress measurement range of 0–14122 kPa.

The current response–strain curve of the HMFPS-10 is shown in Appendix A. The current variation rate of the HMFPS increases with the increase in the strain. The current variation rate of the HMFPS increases relatively slowly when the strain is less than 76%, while the current variation rate of the HMFPS increases rapidly when the strain is more than 76%. This is related to the porosity of the GF sensing layer. With the compressing deformation of the GF, the foam skeletons contact each other and establish new conductive paths, resulting in the current increasing. When the compressing deformation is smaller than the range of the GF porosity, the contact between the foam skeleton is limited, the new conductive path is less, and the current increases slowly. When the compressing deformation is close to or greater than the porosity of the GF, the deformation leads to direct contact of the foam skeleton, resulting in a large number of conductive paths and a sharp increase in the current.

In order to study the dynamic performance of the sensor, we tested the dynamic response of the sensor under 20%, 40%, 60% and 80% compressive strain, as shown in Figure 5d. The response of the sensor can be clearly distinguished with increasing stress, and the three loading cycles under the same stress exhibit good repeatability. In addition, the dynamic responses of the sensor at four different loading rates of 10, 20, 40, and 60 mm min^−1^ under 60% strain are shown in Figure 5e. The results show that the response of the sensor is independent for different compression speeds. Figure 5f shows the response and recovery time of the HMFPS. The HMFPS has a fast response speed to the applied stress, and its response time and recovery time are 83 ms and 166 ms, respectively.

Repeatability and stability are important indicators in the practical application of stress sensors. In order to investigate the cyclic repeatability and stability of the HMFPS, 2000 cyclic loading/unloading tests were performed on the HMFPS, as shown in Figure 5g. The current variation rate of the sample remains stable during multiple cycles. The illustration in Figure 5g shows that the current variation is almost the same for each cycle. Experiments show that our proposed HMFPS has good repeatability and stability. Furthermore, we compared the performance of this work with existing literature on piezoresistive stress sensors, as shown in Figure 5h and Appendix A [17,18,37,45,46,47,48,49,50,51,52,53,54,55,56,57,58,59,60,61,62]. Benefiting from the HM design of the sensor, the prepared HMFPS has high sensitivity and wide measurement range.

### 3.5. Applications of the HMFPS

In order to verify the applicability of the HMFPS in human motion detection, the fabricated HMFPS was attached to different parts of the human body with breathable medical bandage. Additionally, the upper surface of the sensor is covered with the same size paper. Figure 6a is the current response curve of the finger pressing the HMFPS, which shows that the HMFPS can respond to the stress in real time and the current variation rate is proportional to the compressive stress. Figure 6b–f shows the response results of the sensors fixed on the finger, the neck, the elbow, the wrist, and the knee, respectively. The results show that the HMFPS can respond to the motion of the human body through the current variation rate, indicating that the proposed sensor can be used for human motion detection.

## 4. Conclusions

In summary, we propose a HMFPS with high sensitivity and wide measurement range through HM design. The HMFPS consists of a GF sensing layer, a PDMS supporting layer and an interdigital electrode. We prepared three HMFPS with different area (HMFPS-3, HMFPS-5, HMFPS-10), and investigated the effect and mechanism of the PDMS supporting layer area on the piezoresistivity of the sensor. Moreover, we analyzed the sensing mechanism of the HMFPS, and proved that the HM design is an effective method to achieve high sensitivity and a wide measurement range. Among them, HMFPS-10 has the best overall performance, with a sensitivity of 0.695 kPa^−1^, a measurement range of 0–14,122 kPa, fast response/recovery (83 ms and 166 ms) and excellent stability (2000 cycles). In addition, the application of the HMFPS in human motion monitoring was demonstrated.

## Figures and Tables

**Figure 1 micromachines-14-00716-f001:**
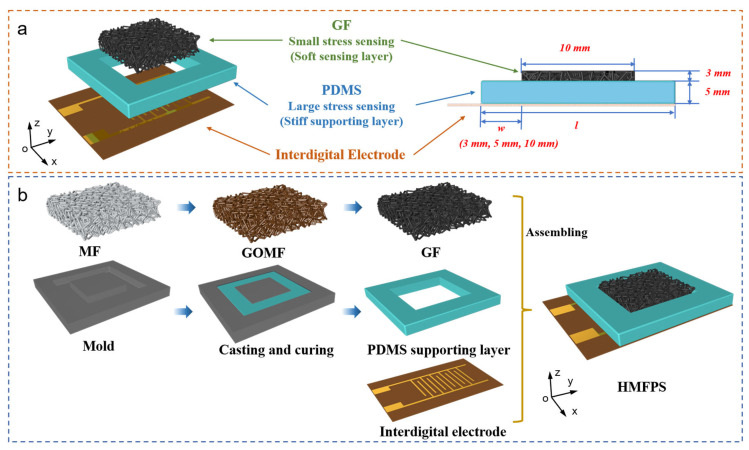
(**a**) Schematic of the HMFPS; (**b**) Schematic of the fabrication process of the HMFPS.

**Figure 2 micromachines-14-00716-f002:**
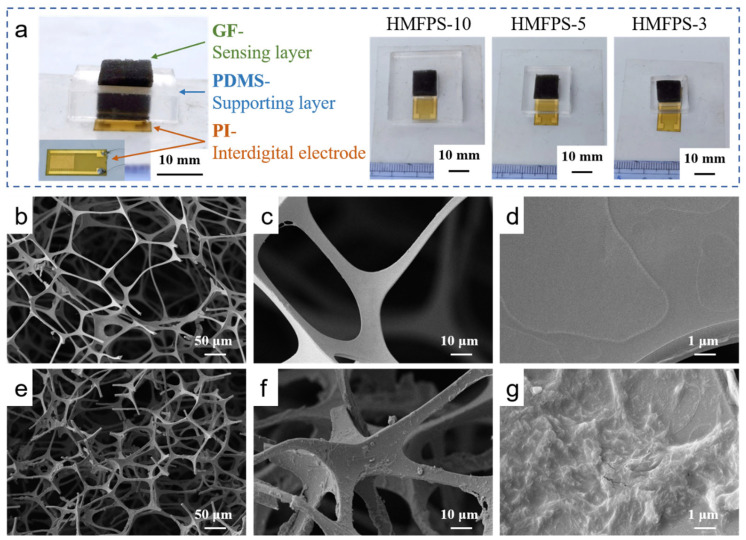
(**a**) Photographs of the HMFPS-3, HMFPS-5 and HMFPS-10; SEM images of (**b**–**d**) the MF and (**e**–**g**) the GF.

**Figure 3 micromachines-14-00716-f003:**
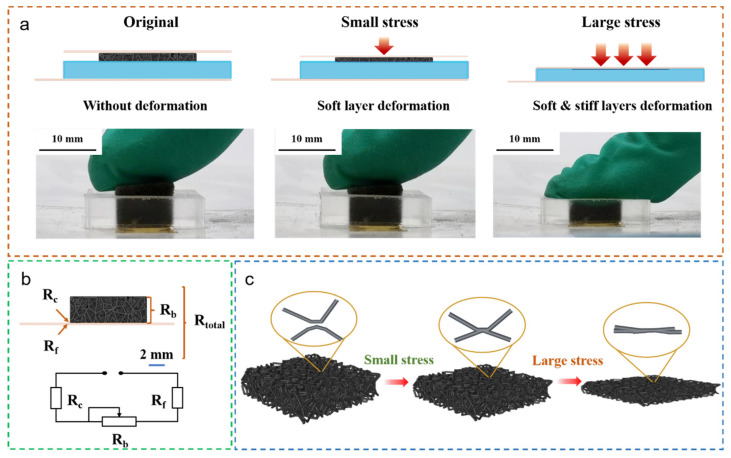
(**a**) Schematic and pictures of different amounts of compressive strain of HMFPS; (**b**) The equivalent circuit model of the HMFPS; (**c**) Schematics of the sensing mechanism of the HMFPS.

**Figure 4 micromachines-14-00716-f004:**
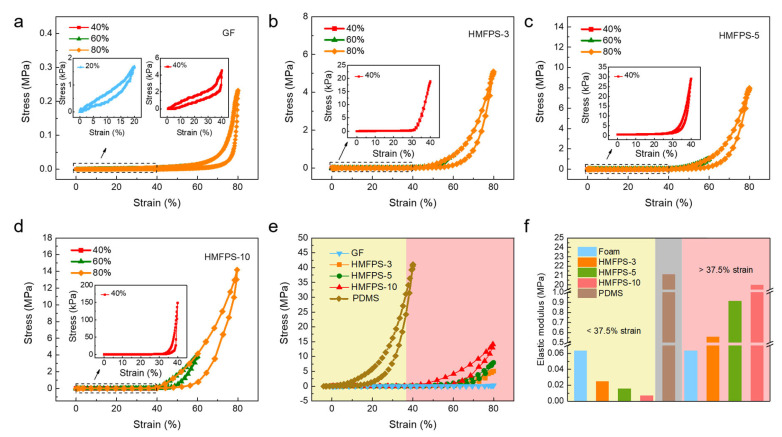
Compressive stress–strain curves of (**a**) GF, (**b**) HMFPS-3, (**c**) HMFPS-5, and (**d**) HMFPS-10 under 40%, 60%, and 80% strain; (**e**) Compressive stress–strain curves, and (**f**) elastic modulus of Gf, HMFPS-3, HMFPS-5, HMFPS-10, and PDMS bulk.

**Figure 5 micromachines-14-00716-f005:**
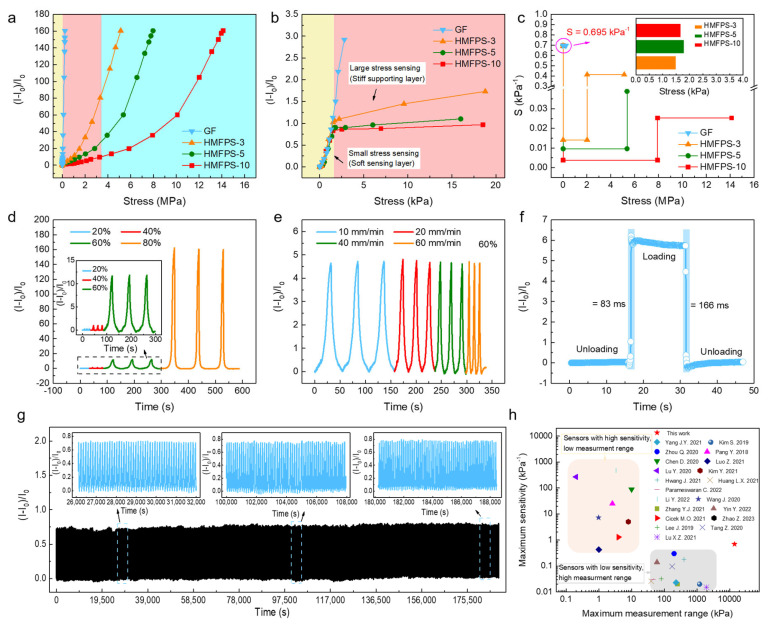
(**a**,**b**) Current responses of the Gf, HMFPS-3, HMFPS-5, and HMFPS-10; (**c**) Sensitivity and detection range of the Gf, HMFPS-3, HMFPS-5, and HMFPS-10; Current variation rates of the HMFPS-3 to cyclic compression (**d**) from 20% strain to 80% strain and (**e**) to different stress rates; (**f**) Response and recovery time of the HMFPS-3; (**g**) Durability test of the HMFPS over 2000 compression cycles; (**h**) Comparison of this sensor with state-of-the-art counterparts.

**Figure 6 micromachines-14-00716-f006:**
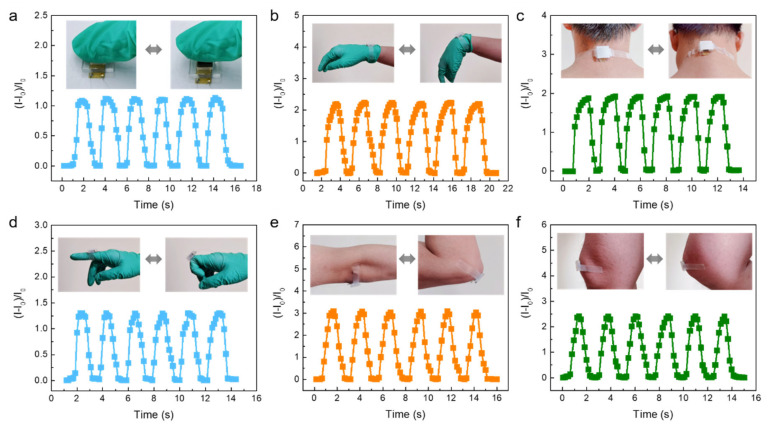
Monitoring human motion using HMFPS; (**a**) Finger pressing, (**b**) wrist bending, (**c**) nodding, (**d**) finger movement, (**e**) arm bending, and (**f**) knee bending.

## Data Availability

Data available on request from the authors.

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
