# Peer review of "Heterogeneous Multi-Material Flexible Piezoresistive Sensor with High Sensitivity and Wide Measurement Range"

_micromachines, 2023, doi:10.3390/mi14040716_

Round 1
Reviewer 1 Report
This paper presents an approach to fabricate and characterize a heterogeneous multi-mate- 20 rial flexible piezoresistive sensor. This sensor has advantages of regular flexible piezoresistive sensors. In addition to them, it has high sensitivity and wide measurement range. Sensors with three different sizes were fabricated and imperially characterized. A wearable applications for different parts of the human body were presented. The sensor has a cover result as well as an interesting fabrication process. It is easy to read and understand motivation of every step. I pointed out some minor comments that would straighten the current draft. Please, address them in a minor revision.
Page 1 line 16: Flexible piezoresistive sensors (FPSs) are widely used in motion detection, wearable devices, and electronic skins due to simple structure, convenient signal acquisition and fast dynamic response.
Page 1 line 22: PDMS layer probably? Or PDMS membrane?
Page 1 line 27: What does HMFPS-10 mean? How does it differ from HMFPS introduced before?
Page 1 line 35: Please, add a relevant reference for a stretchable strain sensor: Shintake, Advanced Material Technologies, 2017,“Ultrastretchable Strain Sensors Using Carbon Black-Filled Elastomer Composites and Comparison of Capacitive Versus Resistive Sensors”
Page 2 line 47: Please, introduce piezoresistive material in the main text before using PM.
Page 2 line 52: CFs have the advantage 52 of high sensitivity. - Is there a reference to back it up?
Page 2 line 80: Throughout the paper you never mentioned why you need an interdigital electrode structure. I suggest to add a clear explanation why it is needed.
Page 2 line 82: What do all the proposed names mean? It is important to explain in the text what HMFPS-3,5,10 stands for. As early as possible.
Page 4 line 135: Thank you for the clear explanation and motivation of SEM images.
Page 7 line 248: The graph is clear. I have a question about the relationship between compressive strain and dynamic response. As we can see from the graph 5d, it is non-linear: the response for 80% and 60% vary significantly. Is there a model to determine such a behaviour?
Page 8 line 264: An application is great and the demonstration is clear. It would be great to see the comparison between your sensor and SoA examples for the specific application - wearable sensor. It is a minor suggestion, which would increase more value of the paper.
Author Response
Dear reviewers:
Thank you for your thoughtful and thorough comments on our manuscript. We have carefully considered all the comments and made necessary corrections in the revised manuscript. Below we provide an item–by–item reply to your comments.
The points raised by you are shown in black whereas our response is written in blue and the content in the manuscript is in red font. The revised part in the manuscript is in a black font with highlighted in yellow. The line numbers in the responses refer to the revised manuscript.
Reviewer 1: This paper presents an approach to fabricate and characterize a heterogeneous multi-material flexible piezoresistive sensor. This sensor has advantages of regular flexible piezoresistive sensors. In addition to them, it has high sensitivity and wide measurement range. Sensors with three different sizes were fabricated and imperially characterized. Wearable applications for different parts of the human body were presented. The sensor has a cover result as well as an interesting fabrication process. It is easy to read and understand motivation of every step. I pointed out some minor comments that would straighten the current draft. Please, address them in a minor revision.
Response:
We thank the reviewer for spending time evaluating the manuscript and providing constructive comments to improve the manuscript.
Page 1 line 16: Flexible piezoresistive sensors (FPSs) are widely used in motion detection, wearable devices, and electronic skins due to simple structure, convenient signal acquisition and fast dynamic response.
Response:
Thank you for the helpful suggestion. We have revised this sentence in the manuscript. (See Paragraph 1 line 16)
Page 1 line 22: PDMS layer probably? Or PDMS membrane?
Response:
Thank you for the helpful comment. PDMS here refers to the PDMS layer, which was modified in the manuscript. (See Paragraph 1 line 22)
Page 1 line 27: What does HMFPS-10 mean? How does it differ from HMFPS introduced before?
Response:
Thank you for the helpful suggestion. HMFPS-10 means a heterogeneous multi-material flexible piezoresistive sensor with the PDMS supporting layers width of 10 mm ( ). And we have modified Figure 1 in the manuscript for clearer representation.
Figure 1. (a) Schematic of the HMFPS; (b) Schematic of the fabrication process of the HMFPS.
(See Figure 1 in Manuscript)
Page 1 line 35: Please, add a relevant reference for a stretchable strain sensor: Shintake, Advanced Material Technologies, 2017,“Ultrastretchable Strain Sensors Using Carbon Black-Filled Elastomer Composites and Comparison of Capacitive Versus Resistive Sensors”
Response:
Thank you very much for your time and thoughtful comments. We have referenced the articles mentioned about a stretchable strain sensor.
Flexible stress sensors are flexible electronics that conformal to curved surfaces and convert stress into electrical signals[1-4], which are widely used in motion detection[5-7], wearable electronics[8, 9], human-computer interaction[10-12], and electronic skin[13-15].
(See Ref 7 in the manuscript)
Page 2 line 47: Please, introduce piezoresistive material in the main text before using PM.
Response:
Thank you for the helpful suggestion. We have revised in the manuscript, introducing piezoresistive material in the manuscript before using PM. (See Paragraph 2 line 47)
Page 2 line 52: CFs have the advantage of high sensitivity. - Is there a reference to back it up?
Response:
Thank you for the helpful suggestion. We have referenced the relevant literature to support this conclusion that CFs have the advantage of high sensitivity. And modified in the article.
Detail revised in the manuscript:
“CFs have the advantage of high sensitivity. While the measurement range of CFs is limited due to the low deformability.[32]”
(See Paragraph 2 line 53)
Page 2 line 80: Throughout the paper you never mentioned why you need an interdigital electrode structure. I suggest to add a clear explanation why it is needed.
Response:
Thank you for the helpful suggestion. The high sensitivity of the HMFPS prepared in this study is achieved by applying initial stress only to the intermediate GF sensing layer. The interdigital electrode is a planar structure, which can be directly fixed on the lower surface of the sensor, so that the prepared sample structure is compact. At the same time, the interdigital electrode can avoid exposed wire connections, which can make the sensor obtain better stability. However, the conventional electrode structure needs to be fixed to the upper and lower surfaces of the sensing foam, forming a sandwich structure that requires a separate lead wire connection. This will increase the complexity of the sensor in connection and structure and affect the stability of the sensor. Therefore, our investigate selected the interdigital electrode as the electrode of the sensor.
Detail added in the manuscript:
“The interdigital electrode enables the sensor to be compact and avoid exposed wire connections, increasing the stability of the sensor.”
(See Paragraph 3 line 116)
Page 2 line 82: What do all the proposed names mean? It is important to explain in the text what HMFPS-3,5,10 stands for. As early as possible.
Response:
Thank you for the helpful suggestion. The HMFPS-3, HMFPS-5 and HMFPS-10 mean the heterogeneous multi-material flexible piezoresistive sensor with different PDMS supporting layers width of 3 mm, 5 mm and 10 mm ( ), respectively. And we have modified Figure 1 in the manuscript for clearer representation.
Detail revised in the manuscript:
“We prepared three different sizes of PDMS supporting layers with widths of 3 mm, 5 mm, and 10 mm, respectively. Finally, the GF, PDMS supporting layer and interdigital electrode were assembled to obtain the HMFPS. The GF is fixed on the surface of the interdigital electrode using a conductive silver paste. And the copper wire is fixed to the interdigital electrode by soldering tin. The HMFPSs with three different supporting layer sizes are defined as HMFPS-3, HMFPS-5, and HMFPS-10, respectively.”
(See Paragraph 3 line 112)
Figure 1. (a) Schematic of the HMFPS; (b) Schematic of the fabrication process of the HMFPS.
(See Figure 1 in Manuscript)
Page 4 line 135: Thank you for the clear explanation and motivation of SEM images.
Response:
Thank you for the helpful suggestion. SEM can observe the morphology of the GF. On the one hand, we can know that the dip-coating method can preserves the original pore structure of the MF. On the other hand, we can prove that graphene sheets are successfully assembled on the MF skeleton.
Page 7 line 248: The graph is clear. I have a question about the relationship between compressive strain and dynamic response. As we can see from the graph 5d, it is non-linear: the response for 80% and 60% vary significantly. Is there a model to determine such a behavior?
Response:
Thank you for the helpful suggestion. The current response of the sensor is nonlinear. To illustrate the sensor's current response at 80% and 60%, we supplement the HMFPS’s current variation rate-strain curve (Figure S3). It can be clearly seen that the current variation rate of the HMFPS-10 increases with the increase of the strain, and the variation is nonlinear. The current response-strain curve of the HMFPS-10 is shown in Figure S3. The current variation rate of the HMFPS increases with the increase of the strain. The current variation rate of the HMFPS increases relatively slowly when the strain is less than 76%, while the current variation rate of the HMFPS increases rapidly when the strain is more than 76%. This is related to the porosity of the GF sensing layer. With the compressing deformation of the GF, the foam skeleton contacts with each other and establishes new conductive paths, resulting in current increasing. When the compressing deformation is smaller than the range of the GF porosity, the contact between the foam skeleton is limited, the new conductive path is less, and the current increases slowly. When the compressing deformation is close to or greater than the porosity of the GF, the deformation leads to direct contact of the foam skeleton, resulting in a large number of conductive paths and a sharp increase in the current.
In addition, we added the foam porosity test in the manuscript. The experimental results show that the porosity of the foam is 78.62%, which is consistent with the sensing performance.
Detail added in the manuscript:
“The porosity of the GF can be expressed as , where represents the volume of GF in its natural state ( ); represents the porous volume of GF). We tested the porosity of the GF by drainage method. The quality of the GF is 6.75 mg. The density of the GF can be calculated as 8.43 kg m-3. And when the GF is full of water, its quality is 635.71 mg. Therefore, the quality of water absorbed by GF pores is 628.96 mg, which is equivalent to the volume of GF pores ( ). Hence, the porosity of the GF is 78.62%.”
(See Paragraph 4 line 148)
Detail added in the manuscript:
“The current response-strain curve of the HMFPS-10 is shown in Figure S3. The current variation rate of the HMFPS increases with the increase of the strain. The current variation rate of the HMFPS increases relatively slowly when the strain is less than 76%, while the current variation rate of the HMFPS increases rapidly when the strain is more than 76%. This is related to the porosity of the GF sensing layer. With the compressing deformation of the GF, the foam skeleton contacts with each other and establishes new conductive paths, resulting in current increasing. When the compressing deformation is smaller than the range of the GF porosity, the contact between the foam skeleton is limited, the new conductive path is less, and the current increases slowly. When the compressing deformation is close to or greater than the porosity of the GF, the deformation leads to direct contact of the foam skeleton, resulting in a large number of conductive paths and a sharp increase in the current.”
(See Paragraph 8 line 270)
Figure S3. Current responses-strain curves of the HMFPS-10.
(See Figure S3 in supporting information)
Page 8 line 264: An application is great and the demonstration is clear. It would be great to see the comparison between your sensor and SoA examples for the specific application - wearable sensor. It is a minor suggestion, which would increase more value of the paper.
Response:
Thank you for the helpful suggestion. In order to illustrate that the HMFPS proposed in this study has excellent sensing performance, high sensitivity and wide measurement range. We compared the performance of this work with existing literature on piezoresistive stress sensors. And the data is supplemented in supporting information. Benefiting from the HM design of the sensor, the prepared HMFPS has high sensitivity and wide measurement range.
Detail added in the manuscript:
“Furthermore, we compared the performance of this work with existing literature on piezoresistive stress sensors, as shown in Figure 5 (h) and Table S1.”
(See Paragraph 9 line 304)
Table 1. Comparison between the results of the proposed foam and its counterparts.
|
Material |
Maximum sensitivity (kPa-1) |
Maximum measurement range (kPa) |
Ref. |
|
GF/PDMS |
0.695 |
14122 |
This work |
|
Magnetic tilt micropillar arraystructured PDMS membrane |
0.313 |
200 |
[1] |
|
Graphene pressure sensor with random distribution spinosum |
25.1 |
2.6 |
[2] |
|
Unsymmetrical alveolate PMMA/MWCNTs 1 film |
88 |
10 |
[3] |
|
Interlocked ordered nanocone array pressure sensor |
268.36 |
0.2 |
[4] |
|
Carbon nanotube network-coated porous elastomer sponges |
0.02 |
1200 |
[5] |
|
Hydrophilic hierarchical porous PDMS |
0.03 |
45 |
[6] |
|
Liquid metal modulated nitrogen-doped graphene nanosheets sponge |
476 |
3.4 |
[7] |
|
Quasi-hemispherical micropatterned array on the SWCNTs/TPU 2 film |
0.02 |
254.8 |
[8] |
|
Passive particle jamming variable stiffness material-based flexible capacitive stress sensor |
0.023 |
230 |
[9] |
|
Porous expandable polyethylene/loofah-like polyurethane sponge material |
0.000195 |
3000 |
[10] |
|
CNT/SiNPs 3 three-dimensional (3D) printing flexible pressure sensors |
0.096 |
175 |
[11] |
|
Flexible conductive rGO/TPU foam |
0.0152 |
1940 |
[12] |
1 PMMA/MWCNTs: polymethyl methacrylate (PMMA)/multiwalled carbon nanotubes (MWCNTs). 2 SWCNTs/TPU: single-walled carbon nanotubes (SWCNTs)/thermoplastic polyurethane (TPU). 3 CNT/SiNPs: insulating SiNP carbon nanotubes (CNTs) and fumed silica nanoparticles (SiNPs). 4 rGO/TPU: reduced graphene oxide (rGO)/thermoplastic polyurethane (TPU) porous foam.
(See Table S1 in supporting information)

Reviewer 2 Report
In view of the shortcomings of existing piezoresistive materials for flexible pressure sensors, a new type of heterogeneous multi-material piezoresistive material is proposed. The heterogeneous multi-material structure design can make the sensor with high sensitivity and wide measurement range. This research has enough innovation and can effectively improve the performance of the sensor. The author investigated the effects and the principle of heterogeneous multi-material structures on piezoresistive properties of the sensor. It is proved that heterogeneous multi-material structure is an effective way to realize flexible sensor with high sensitivity and wide measuring range. This study provides a new way to improve the performance of flexible strain sensor and has important significance. The manuscript meets the requirement of Micromachines journal for publication, it is suggested to modify the following problems before agreeing to publication:
1. What is the conductivity of GF sensing layer in the process of sensor preparation? And whether the conductivity of GF can be adjusted and improved
2. In Figure 4, please adjust the size of the illustration to ensure that the picture is legible.
3. In this paper, GF was prepared by dip-coating method. How to determine the successful preparation of the GF?
Author Response
Dear reviewers:
Thank you for your thoughtful and thorough comments on our manuscript. We have carefully considered all the comments and made necessary corrections in the revised manuscript. Below we provide an item–by–item reply to your comments.
The points raised by you are shown in black whereas our response is written in blue and the content in the manuscript is in red font. The revised part in the manuscript is in a black font with highlighted in yellow. The line numbers in the responses refer to the revised manuscript.
Reviewer 2: In view of the shortcomings of existing piezoresistive materials for flexible pressure sensors, a new type of heterogeneous multi-material piezoresistive material is proposed. The heterogeneous multi-material structure design can make the sensor with high sensitivity and wide measurement range. This research has enough innovation and can effectively improve the performance of the sensor. The author investigated the effects and the principle of heterogeneous multi-material structures on piezoresistive properties of the sensor. It is proved that heterogeneous multi-material structure is an effective way to realize flexible sensor with high sensitivity and wide measuring range. This study provides a new way to improve the performance of flexible strain sensor and has important significance. The manuscript meets the requirement of Micromachines journal for publication, it is suggested to modify the following problems before agreeing to publication:
Response:
We thank the reviewer for spending time evaluating the manuscript and providing constructive comments to improve the manuscript.
- What is the conductivity of GF sensing layer in the process of sensor preparation? And whether the conductivity of GF can be adjusted and improved.
Response:
Thank you for the helpful suggestion. The GF was prepared by dip-coating method. The conductivity of the GF was related to the content of conductive filler (graphene) adsorbed on the surface of the foam. We prepared five groups of the GF samples with different dip-coating times, and tested their electrical conductivity. The experimental results show that the conductivity of the GF increases with the increase of dip-coating times. The conductivity of the first three dip-coating times increased rapidly, and the conductivity of the fourth and fifth dip-coating times increased slowly. According to the threshold effect, higher sensitivity can be obtained by keeping the conductivity of piezoresistive composites near the threshold value. Therefore, in order to obtain better piezoresistive characteristics, the GF with the largest change in conductivity should be selected. The conductivity of GF3 with the largest conductivity change is about 0.086 S m−1. In the follow-up studies in this manuscript, GF3 samples are all selected.
Detail added in the manuscript:
“We prepared five groups of GOMFs with different dip-coating times, respectively, and chemical reduced to prepare the GF, expressed as GFx (where x = 1, 2, 3, 4, and 5).”
(See Paragraph 3 line 107)
Detail added in the manuscript:
“In this study, five groups of GF samples were prepared by changing the dip-coating times. The conductivity of GF increases with the increase of dip-coating times, as shown in Figure S1. The conductivity of the GF increased rapidly in the first three dip-coating times, while increased slowly in the fourth and fifth dip-coating times. According to the threshold effect, higher sensitivity can be obtained by keeping the conductivity of piezoresistive composites near the threshold value. Therefore, in order to obtain better pressure-sensitive characteristics, GF3 foam samples with the largest change in conductivity should be selected. The conductivity of GF3 is about 0.086 S m−1. In the follow-up studies of this experiment, GF3 samples were used.”
(See Paragraph 5 line 189)
- In Figure 4, please adjust the size of the illustration to ensure that the picture is legible.
Response:
Thank you for the helpful suggestion. We have modified the Figure 4.
Figure 4. Compressive stress-strain curves of (a) GF, (b) HMFPS-3, (c) HMFPS-5, and (d) HMFPS-10 under 40%, 60%, and 80% strain; (e) Compressive stress–strain curves, and (f) Elastic modulus of Gf, HMFPS-3, HMFPS-5, HMFPS-10, and PDMS bulk.
(See Figure 4 in Manuscript)
- In this paper, GF was prepared by dip-coating method. How to determine the successful preparation of the GF?
Response:
Thank you for the helpful suggestion. The GF sample is prepared by fixing the GO on MF surface and reduced the GO to G by chemical reduction. In order to prove the successful preparation of GF, we used Raman spectroscopy to characterize the GF. Figure S2 shows the Raman spectra of the GOMF and the GF contain D and G valence bands at 1343 cm–1 and 1580 cm–1, respectively. The D band is attributed to sp3 defects in the graphene sheet, and the G band is attributed to the E2g phonon mode in the sp2 carbon atoms plane. The ratio of the intensity of the D valence band to the G valence band (ID/IG) of the GOMF was 0.82. The ID/IG of GF was 1.28 after chemical reduction by VC, indicating that the GO sheets attached to MF are efficiently reduced to G. The GF was successfully prepared.
Detail added in the manuscript:
“Raman spectrums were recorded using a Raman spectrometer (Jobin Yvon LavRam HR800, Horiba, JP) with laser excitation of 514 nm.”
(See Paragraph 3 of line 125)
“Figure S2 shows the Raman spectra of the GOMF and the GF contain D and G valence bands at 1343 cm–1 and 1580 cm–1, respectively. The D band is attributed to sp3 defects in the graphene sheet, and the G band is attributed to the E2g phonon mode in the sp2 carbon atoms plane. The ratio of the intensity of the D valence band to the G valence band (ID/IG) of the GOMF was 0.82. The ID/IG of GF was 1.28 after chemical reduction by VC, indicating that the GO sheets attached to MF are efficiently reduced to G.”
(See Paragraph 4 of line 155)
Figure S2. Raman spectrums of the GOMF and the GF.
(See Figure S2 in supporting information)
Reviewer 3: To solve the problem that the flexible piezoresistive sensors based on a single piezoresistive material cannot achieve high sensitivity and wide measurement range simultaneously, a heterogeneous multi-material flexible piezoresistive sensor (HMFPS) with high sensitivity and wide measuring range was designed. This research has clear innovation points and sufficient research significance. The author investigated the influence of the size of the supporting layer on the performance of the sensor, obtaining the sensor samples with good performance. Therefore, the HMFPS can be used for human motion detection. The manuscript meets the Micromachines Journal publication requirements, recommended changes to the following issues before agreeing to publication:
Response:
We thank the reviewer for spending time evaluating the manuscript and providing constructive comments to improve the manuscript.
- The HMFPS can obtain a high sensitivity of 0.695 kPa-1in the small stress range, which is generated by the compression of only the intermediate GF sensing layer. However, according to Figure 5 (c), three sensors of different sizes produce different stress ranges of high sensitivity. Please explain the reasons.
Response:
Thank you for your questions. In the small stress range (< 1.8 kPa), three samples with different sizes have the same high sensitivity (0.695 kPa-1). However, there is a slight difference in the stress range of the three sensors to obtain high sensitivity, which is HMFPS-3 (< 1.5 kPa), HMFPS-5 (< 1.8 kPa), and HMFPS-10 (< 1.669 kPa). This slight difference is caused by irreversible deformation of the GF pressing process. In the process of only pressing GF, the sensor is very sensitive to the stress response, resulting in a small difference in stress.
- Please explain how to fix the sensor on the human body when detecting human motion?
Response:
Thank you for the helpful suggestion. The HMFPS should be fixed in different parts of the human body when detecting human motion. We covered the upper surface of the sensor with paper of the same size and fixed it on the human body with breathable medical bandage. The purpose of covering the paper is to ensure that the force area of the sensor is consistent throughout the measurement process. During the experimental test, the area of the sensor is also equivalent to the area of the whole sensor.
Detail added in the manuscript:
“, the fabricated HMFPS was attached to different parts of the human body with breathable medical bandage. And the upper surface of the sensor is covered with the same size paper.”
(See Paragraph 9 of line 308)
- Please explain how is the copper wire fixed with the interdigital electrode in the sensor test process?
Response:
Thank you for the helpful suggestion. The HMFPS was assembled with the GF, PDMS supporting layer and interdigital electrode. The GF is fixed on the surface of the interdigital electrode using a conductive silver paste. And the copper wire is fixed to the interdigital electrode by soldering tin.
Detail added in the manuscript:
“The GF is fixed on the surface of the interdigital electrode using a conductive silver paste. And the copper wire is fixed to the interdigital electrode by soldering tin.”
(See Paragraph 3 of line 114)

Reviewer 3 Report
To solve the problem that the flexible piezoresistive sensors based on a single piezoresistive material cannot achieve high sensitivity and wide measurement range simultaneously, a heterogeneous multi-material flexible piezoresistive sensor (HMFPS) with high sensitivity and wide measuring range was designed. This research has clear innovation points and sufficient research significance. The author investigated the influence of the size of the supporting layer on the performance of the sensor, obtaining the sensor samples with good performance. Therefore, the HMFPS can be used for human motion detection. The manuscript meets the Micromachines Journal publication requirements, recommended changes to the following issues before agreeing to publication:
1. The HMFPS can obtain a high sensitivity of 0.695 kPa-1 in the small stress range, which is generated by the compression of only the intermediate GF sensing layer. However, according to Figure 5 (c), three sensors of different sizes produce different stress ranges of high sensitivity. Please explain the reasons.
2. Please explain how to fix the sensor on the human body when detecting human motion?
3. Please explain how is the copper wire fixed with the interdigital electrode in the sensor test process?
Author Response
Dear reviewers:
Thank you for your thoughtful and thorough comments on our manuscript. We have carefully considered all the comments and made necessary corrections in the revised manuscript. Below we provide an item–by–item reply to your comments.
The points raised by you are shown in black whereas our response is written in blue and the content in the manuscript is in red font. The revised part in the manuscript is in a black font with highlighted in yellow. The line numbers in the responses refer to the revised manuscript.
Reviewer 3: To solve the problem that the flexible piezoresistive sensors based on a single piezoresistive material cannot achieve high sensitivity and wide measurement range simultaneously, a heterogeneous multi-material flexible piezoresistive sensor (HMFPS) with high sensitivity and wide measuring range was designed. This research has clear innovation points and sufficient research significance. The author investigated the influence of the size of the supporting layer on the performance of the sensor, obtaining the sensor samples with good performance. Therefore, the HMFPS can be used for human motion detection. The manuscript meets the Micromachines Journal publication requirements, recommended changes to the following issues before agreeing to publication:
Response:
We thank the reviewer for spending time evaluating the manuscript and providing constructive comments to improve the manuscript.
- The HMFPS can obtain a high sensitivity of 0.695 kPa-1in the small stress range, which is generated by the compression of only the intermediate GF sensing layer. However, according to Figure 5 (c), three sensors of different sizes produce different stress ranges of high sensitivity. Please explain the reasons.
Response:
Thank you for your questions. In the small stress range (< 1.8 kPa), three samples with different sizes have the same high sensitivity (0.695 kPa-1). However, there is a slight difference in the stress range of the three sensors to obtain high sensitivity, which is HMFPS-3 (< 1.5 kPa), HMFPS-5 (< 1.8 kPa), and HMFPS-10 (< 1.669 kPa). This slight difference is caused by irreversible deformation of the GF pressing process. In the process of only pressing GF, the sensor is very sensitive to the stress response, resulting in a small difference in stress.
- Please explain how to fix the sensor on the human body when detecting human motion?
Response:
Thank you for the helpful suggestion. The HMFPS should be fixed in different parts of the human body when detecting human motion. We covered the upper surface of the sensor with paper of the same size and fixed it on the human body with breathable medical bandage. The purpose of covering the paper is to ensure that the force area of the sensor is consistent throughout the measurement process. During the experimental test, the area of the sensor is also equivalent to the area of the whole sensor.
Detail added in the manuscript:
“, the fabricated HMFPS was attached to different parts of the human body with breathable medical bandage. And the upper surface of the sensor is covered with the same size paper.”
(See Paragraph 9 of line 308)
- Please explain how is the copper wire fixed with the interdigital electrode in the sensor test process?
Response:
Thank you for the helpful suggestion. The HMFPS was assembled with the GF, PDMS supporting layer and interdigital electrode. The GF is fixed on the surface of the interdigital electrode using a conductive silver paste. And the copper wire is fixed to the interdigital electrode by soldering tin.
Detail added in the manuscript:
“The GF is fixed on the surface of the interdigital electrode using a conductive silver paste. And the copper wire is fixed to the interdigital electrode by soldering tin.”
(See Paragraph 3 of line 114)
